# Position: Safety Must Precede the Deployment of Open-Ended AI Agents

Ivaxi Sheth [1]   Jan Wehner [* 1]   Sahar Abdelnabi [* 2]   Ruta Binkyte [* 1]   Mario Fritz [1]

## Abstract

AI advancements have been significantly driven by a combination of foundation models and curiosity-driven learning aimed at increasing capability and adaptability. Within this landscape, open-endedness, where AI agents autonomously and indefinitely generate novel behaviors, representations, or solutions, has gained increasing interest. This has become relevant in the context of self-evolving agents and long-horizon discovery. **This position paper argues that the defining properties of open-ended AI systems introduce a distinct and underexplored class of safety challenges, including loss of predictability, emergent misalignment, and difficulties in maintaining effective control as systems evolve beyond their initial design assumptions, that must be addressed preemptively.** These challenges differ qualitatively from those associated with task-bounded or static models and are unlikely to be addressed by existing safety frameworks alone, which is why these risks must be examined proactively, before large-scale deployment. The paper proposes a taxonomy for key challenges, discusses research opportunities, and calls for coordinated action to support the safe and responsible development of open-ended AI.

## 1. Introduction

Artificial Intelligence (AI) has achieved remarkable progress driven by foundation models (Bommasani et al., 2021; Ramesh et al., 2021; Rombach et al., 2022; Achiam et al., 2023; Radford et al., 2023; Brooks et al., 2024). Beyond static prediction and generation, recent systems increasingly use foundation models as components in agentic and self-evolving architectures, enabling autonomous exploration, tool use, self-reflection, and iterative improvement over extended horizons (Zhang et al., 2025; OpenClaw, 2026; Nous Research, 2026). These developments move AI systems closer to open-ended discovery. Open-ended discovery is key to making progress on problems that cannot be solved by following a specified objective (Stanley and Lehman, 2015). Indeed, humans use such open-ended processes to accumulate knowledge and solve difficult problems (Land and Hannafin, 1997). Thus, previous work argued that open-endedness is key for Artificial Superintelligence (Stanley, 2019; Team et al., 2021; Jiang et al., 2023; Nisioti et al., 2024; Hughes et al., 2024), which could outperform humans at a range of tasks (Morris et al., 2024).

Open-Ended (OE) AI continuously produces artifacts that are novel and learnable to humans. This enables it to generate new, complex, creative, and adaptive solutions over time (Soros and Stanley, 2014; Soros et al., 2017; Clune, 2019; Sigaud et al., 2023; Lu et al., 2024; Akiba et al., 2025). Unlike non-adaptive AI that optimizes for fixed objectives, OE AI adapts to changing circumstances without being given an explicit goal.

Historically, OE AI has been grounded in evolutionary and quality-diversity methods, which search for increasingly complex, diverse artifacts without relying on a fixed objective (Pugh et al., 2016; Alvarez et al., 2019). Guiding OE AI toward artifacts that are novel and meaningful to humans has been difficult. Recent works use Large Language Models (LLMs) to accelerate this process by leveraging their implicit understanding of human preferences, learned from large-scale data. Hence, this allows LLMs as general-purpose backbones for open-ended evolution and exploration, enabling agents to generate, evaluate, and refine behaviors or artifacts with minimal human intervention (Lehman et al., 2023; Zammit et al., 2024; Aki et al., 2024). This integration substantially broadens the scope of OE AI by combining unbounded exploration with powerful knowledge and reasoning capabilities. As a result, LLM-based OE systems have exhibited emergent behaviors in domains such as scientific discovery (Lu et al., 2024; team, 2025; Tang et al., 2026), long-horizon navigation in previously unseen environments (Wang et al., 2024a), and the generation and interaction with entirely novel environments (Bruce et al., 2024).

*Equal contribution [1]CISPA-Helmholtz Center of Information Security [2]MPI for Intelligent Systems, ELLIS Institute Tübingen, Tübingen AI Center. Correspondence to: Ivaxi Sheth <ivaxi.sheth@cispa.de>.

*Proceedings of the 43$^{rd}$ International Conference on Machine Learning*, Seoul, South Korea. PMLR 306, 2026. Copyright 2026 by the author(s).

There is a wide diversity of systems that realize open-ended learning across agents, environments, and tasks. Many approaches co-evolve agents and environments to generate progressively harder and more diverse challenges while enabling skill transfer across tasks (Wang et al., 2019). Others employ LLM-powered embodied agents that use automated curricula, skill libraries, and self-reflection to achieve continual open-ended learning in dynamic worlds (Wang et al., 2024a; Sarch et al., 2023; Nachkov et al., 2025; Hu et al., 2025; Qu et al., 2026; Zhang et al., 2026a). OE methods have been applied to coding and program synthesis (Zhang et al., 2025), scientific discovery (team, 2025; Yamada et al., 2025; Agarwal et al., 2025; Lange et al., 2025; Fawzi et al., 2022; Romera-Paredes et al., 2024), robotics and sensori-motor skill acquisition (Cartoni et al., 2020; 2023), and the generation of evolving game worlds (Che et al., 2024; Earle et al., 2021; Bruce et al., 2024). More broadly, OE AI has been proposed as a mechanism for agents to autonomously accumulate skills and knowledge over long horizons in rich environments, and as a potential pathway toward Artificial Superintelligence (Team et al., 2021; Hughes et al., 2024; Nisioti et al., 2024).

There is a rapidly growing interest in OE AI as a pathway toward more general and continually improving artificial intelligence; examples of that are recent workshops (neu, 2023; 2024) and ICLR'25 keynote talk (Rocktaeschel, 2025). As OE AI increasingly intersects with agentic autonomy, memory, and tool use, its deployment beyond a sandboxed environment becomes more plausible. This trajectory makes it essential to evaluate and mitigate emerging risks alongside continued capability development.

**Position: While OE AI offers substantial potential benefits, it also poses unique and significant risks that must be addressed preemptively before further development. Its inherent unpredictability and long-horizon dynamics necessitate dedicated safety research.**

While AI safety is a broad and active area of research, this paper focuses on the distinct challenges introduced by open-ended (OE) AI systems. Building on earlier discussions by Hughes et al. (2024) and Ecoffet et al. (2020), which briefly raised safety concerns in open-ended settings, we identify emerging and underexplored risks that arise from continual novelty generation and self-directed adaptation, particularly in LLM-based agentic systems. Although fully open-ended AI systems are not yet widely deployed, recent autonomous agents already exhibit precursor dynamics. In one reported incident, an agent operating over a real email inbox lost or failed to preserve a safety-relevant instruction during long-context execution and began deleting emails despite a prior confirmation constraint (Flynn, 2026). Large-scale agent communities such as MoltBook show behavioral drift and incentive-sensitive adaptation over time (Feng et al., 2026).

Separately, rapidly growing agent ecosystems have exposed security weaknesses in tool and skill registries, illustrating how capability growth can outpace safety infrastructure (CrewClaw, 2026). These examples motivate studying OE risks before such systems become more autonomous, persistent, and widely deployed.

In this paper, we characterize key failure modes, establish conceptual foundations for reasoning about OE-specific risks, and outline research directions spanning empirical evaluation, technical mitigation, and governance. Although our primary focus is on systems that explicitly optimize for novelty and continual adaptation, many adaptive models, such as LLM-based agents with persistent memory (DeChant, 2025; Pulipaka et al., 2026) or increasing autonomy (Tang et al., 2024) exhibit similar dynamics and lie along a continuum from static to fully open-ended systems. Given the growing societal relevance of such systems, we argue that these risks should be addressed early. By outlining potential hazards and candidate mitigation strategies, this paper aims to support the safe development of OE AI and to provide a foundation for future oversight and policy frameworks (Biden, 2023; Edwards, 2021). In summary, the contributions of this position paper are:

- Identifying and proposing a taxonomy for key OE-specific safety risks that are distinct from the current studied risks,

- Outlining mitigation-oriented research directions,

- Discussing implications for responsible development and governance.

## 2. What is Open-Endedness?

There have been multiple definitions for Open-Endedness (Stanley and Soros, 2016; Soros et al., 2017; Stanley and Lehman, 2015; Lehman and Stanley, 2011). One definition frames OE as generating artifacts that are novel and learnable for an external observer (Hughes et al., 2024). This definition introduces subjectivity, as novelty can be evaluated differently depending on the observer, and excludes systems generating unintelligible artifacts (e.g., TV noise). Another view models OE systems via evolutionary principles, prioritizing diversity and incremental complexity in behaviors or solutions (Packard et al., 2019). Such systems autonomously create and solve problems without direct human intervention, mimicking the processes of biological evolution. Another perspective views OE as a search problem characterized by continuous exploration across a vast and evolving state space, generating diverse and increasingly complex solutions without explicit end goals (Sigaud et al., 2023). We adopt the definition by Hughes et al. (Hughes et al., 2024), which frames OE as generating novel and learnable artifacts to an external observer. This is particularly suited for ML

contexts and facilitates a structured approach to identifying risks w.r.t. the observer incurred by the evolving nature.

## 2.1. Definition

*An open-ended AI system continuously generates artifacts that are novel and learnable to an observer.*

Consider a system $S$ that generates a sequence of artifacts $A_{1:\infty}$, indexed by time, where each artifact $A_t$ lies in an artifact space $\mathcal{A}$. Let $\mathcal{H}_t = \mathcal{A}^t$ denote the space of artifact histories up to time $t$. An observer $O$ updates its internal predictive state after observing a history of artifacts. We write this update as

$$M_t := \mathrm{Obs}(A_{1:t}), \tag{1}$$

where $\mathrm{Obs} : \mathcal{H}_t \to \mathcal{M}$ maps the observed history to an observer predictive state $M_t$.

The observer predictive state $M_t$ induces a predictive distribution over future artifacts. For any future time $t' > t$, we denote this prediction by

$$P_{M_t}(A_{t'} \mid A_{1:t}), \tag{2}$$

where the conditioning emphasizes that the prediction is based only on the history available up to time $t$. The observer's prediction quality is measured by a loss function $\mathcal{L}(M_t, A_{t'})$, which compares the observer's prediction after observing $A_{1:t}$ with the realized future artifact $A_{t'}$. Expectations are taken over the randomness of the artifact-generating system $S$, and over observer randomness.

Borrowing from Hughes et al. (Hughes et al., 2024), we consider a system to display **novelty** if it produces artifacts that become progressively less predictable to a fixed observer predictive state. Formally:

$$\forall t < t' \; \exists t^* > t' : \mathbb{E}[\mathcal{L}(M_t, A_{t'})] < \mathbb{E}[\mathcal{L}(M_t, A_{t^*})]. \tag{3}$$

This means that, for an observer whose predictive state is fixed at time $t$, there will always be later artifacts that are harder to predict, ensuring that the system continues to produce novel artifacts.

OE AI is **learnable** if incorporating a longer history of artifacts improves the observer's ability to predict future outputs. This is formalized as:

$$\forall t < t' < t^* : \mathbb{E}[\mathcal{L}(M_t, A_{t^*})] > \mathbb{E}[\mathcal{L}(M_{t'}, A_{t^*})]. \tag{4}$$

Here, the observer predictive state $M_{t'}$ is obtained after seeing a longer history than $M_t$. The decrease in loss indicates that the observer can learn structure in the artifact-generating process over time.

Although our analysis focuses on systems that satisfy a formal definition of open-endedness, many contemporary

AI systems lie along a continuum from static models to fully open-ended agents such as memory-augmented LLMs (Maharana et al., 2024; Xu et al., 2026; Chhikara et al., 2025; Pink et al., 2025; Wu et al., 2026) and self-evolving agents (Gao et al., 2025; Shao et al., 2025; Wu et al., 2025). Several risks discussed in this paper therefore arise from these mechanisms and can also appear, in attenuated form, in partially adaptive systems. In practice, our primary concern is with **agentic** (Andreas, 2022) **OE systems built on LLMs**. For brevity, we refer to this broader class as *open-ended AI*.

## 2.2. Safety of Open-Ended AI

Safety is often defined as the prevention of unacceptable harm, and in AI is commonly formalized through errors or deviations from predefined objectives or constraints (Suyama, 2005; Kafka, 2012). This framing is difficult to apply directly to OE AI systems, since they continually generate novel artifacts beyond prior design specifications, making it hard to enumerate failures in advance. We therefore adopt a risk-management perspective (Leveson, 2012), where safety requires identifying, assessing, and mitigating risks under novelty and uncertainty.

In OE AI, safety is not a one-time property verified at deployment, but a time-dependent property over future artifact trajectories. Let $h(A_{t'}) \geq 0$ denote the harm associated with a future artifact $A_{t'}$. Given the observer predictive state $M_t$, we define the estimated risk at time $t$ as

$$R_t(A_{t'}) := \mathbb{E}[h(A_{t'}) \mid M_t]. \tag{5}$$

An artifact is acceptably safe at time $t$ if $R_t(A_{t'}) \leq \epsilon$ for a threshold $\epsilon$. A failure of temporal safety certification occurs when an artifact appears safe under the current observer state, but later artifacts exceed the acceptable risk threshold:

$$\exists \, t < t' < t^* : R_t(A_{t'}) \leq \epsilon \quad \text{and} \quad R_t(A_{t^*}) > \epsilon. \tag{6}$$

This captures the challenge that artifacts judged safe at one stage may not remain representative of the risks posed by later, more novel artifacts as the system evolves.

**Comparison against general AI safety.** Open-ended AI safety differs from general AI safety in several structural ways. Whereas conventional AI safety typically assumes a fixed objective, bounded task setting, and evaluation of a comparatively static model, OE AI involves continual novelty generation, evolving behavior, and potentially unbounded horizons. As a result, safety in OE systems cannot be reduced to specification error or one-time post-training evaluation; it must instead address process-level risks such as emergent misalignment, reduced traceability, and the need for adaptive oversight and constraints over time. We summarize these distinctions in Table 1.

# 3. Challenges and Risks

OE AI systems raise fundamental challenges that should be examined prior to further development and deployment. Their continual adaptation and complex dynamics can produce behaviors that diverge from intended objectives, including unsafe, unethical, or misaligned outcomes. This section analyzes these challenges and their implications.

## 3.1. Unpredictability

OE AI systems are inherently unpredictable due to their defining drive to generate novel artifacts. As novelty accumulates over time, future outputs increasingly diverge from prior observations, making them harder to anticipate. Consider an OE system $S$ that produces a sequence of scientific discoveries $A \in \mathcal{A}$. While some future artifacts may be benign, others, such as discoveries enabling dangerous biological agents may pose severe risks. Crucially, at an initial time $t$, it is difficult to foresee or assess the safety of which artifacts will be generated at a later time $t'$.

Formally, we assume that lower loss corresponds to higher predictive probability under the observer's model, such that $\mathcal{L}(M_t, A_{t'}) < \mathcal{L}(M_t, A_{t^*})$ implies $P_{M_t}(A_{t'}) > P_{M_t}(A_{t^*})$, where $P_{M_t}(a)$ denotes the probability assigned by model $M_t$ to artifact $a$. This assumption holds for common losses such as cross-entropy. Under this interpretation, the novelty condition (Definition 3) implies that for any observer at time $t$, there will always exist future artifacts that are less predictable:

$$\forall t < t' \ \exists t^* > t' : \mathbb{E}[P_{M_t}(A_{t'})] > \mathbb{E}[P_{M_t}(A_{t^*})].$$

Recently, an OpenClaw agent that autonomously began deleting a user's email inbox after its context-compaction mechanism discarded an earlier safety instruction to confirm before acting when working memory became saturated (Flynn, 2026). The failure was not adversarial; rather, the agent continued pursuing its task objective after losing safety-relevant state, and the user was unable to halt it remotely. This illustrates how long-horizon agentic systems can violate constraints in ways that are difficult to predict in advance and difficult to diagnose while the failure is unfolding.

Thus, unpredictability is not incidental but structurally part of OE, which undermines our ability to anticipate whether future artifact trajectories $\{A_t\}_{t=n}^{\infty}$ will remain safe, complicating systematic risk management. In contrast, traditional reinforcement learning systems optimize a fixed reward function, which provides a basis for predicting that high-reward trajectories are more likely than low-reward ones. OE AI lacks such a stabilizing objective. Moreover, explicit novelty pressures (Lehman and Stanley, 2011) and evolutionary dynamics (Lehman and Stanley, 2010; Dharna et al.,

2022) actively encourage behavioral divergence, further limiting predictability over long horizons.

> **R1:** OE AI systems are intrinsically unpredictable, limiting ability to anticipate harmful outcomes.

## 3.2. Creativity vs. Control

OE AI introduces a fundamental tension between creativity and control in OE search (Ecoffet et al., 2020).

**Lack of Explicit Guidance.** OE AI typically operates without fixed objectives or constraints, allowing it to explore large regions of the state space and generate creative solutions that cannot be reached by specifying desired outcomes. However, this lack of explicit guidance makes it difficult to steer system behavior toward outcomes that are reliably valuable or safe.

**Evolving Model and Environment.** As OE systems acquire new skills and generate new artifacts, both the agent and its environment evolve over time. As a result, guidance or constraints that were appropriate earlier may become ineffective, requiring continual adaptation and undermining long-term control.

> **R2:** OE AI trades control for creativity, requiring continually updated guidance.

## 3.3. Misalignment

Aligning AI systems with human values is a central challenge in AI safety (Hendrycks et al., 2021; Ji et al., 2024). Classical formulations of alignment assume that an AI system optimizes an explicit, human-designed objective, such as a reward function, loss, or utility. In this setting, misalignment arises when the specified objective fails to capture the designer's true intent (Krakovna et al., 2020), or when the system internalizes goals that diverge from the explicit incentives (Shah et al., 2022; Di Langosco et al., 2022).

OE AI fundamentally departs from this setting. OE systems do not optimize a fixed, explicitly specified objective. As a result, alignment in OE AI cannot be understood as matching a learned policy to a predefined reward, but must instead be framed in terms of how values are encoded, amplified, and transformed by the dynamics of the process itself. Designers may incorrectly encode their values in the OE process, leading the system to systematically favor undesired outcomes. Moreover, OE systems may develop intrinsic drives that diverge from those implicit incentives. A canonical analogy is biological evolution: while evolution optimizes inclusive fitness, humans do not intrinsically value inclusive fitness, but instead pursue proxy drives such as sugar consumption that were advantageous.

**Alignment of Evolving Systems.** In contrast to static sys-

tems, the objectives and behaviors of an OE AI can evolve throughout its lifetime. As a result, alignment guarantees or evaluations performed at one point in time may become invalid as the system continues to adapt. This can be expressed in the same loss-based framework as our OE definition, although it is not derivable from novelty and learnability alone because it additionally requires a notion of human-value alignment. Let $\mathcal{L}_{\text{align}}(M_t, A_{t'})$ denote the alignment loss assigned by the observer model $M_t$ to a future artifact $A_{t'}$, where lower values indicate better alignment with human values. An artifact is certified as aligned at time $t$ if

$$\mathbb{E}[\mathcal{L}_{\text{align}}(M_t, A_{t'})] \leq \epsilon$$

for some threshold $\epsilon$. Alignment certification fails if

$$\exists t < t' < t^* : \quad \mathbb{E}[\mathcal{L}_{\text{align}}(M_t, A_{t'})] \leq \epsilon \qquad (7)$$
$$\wedge \quad \mathbb{E}[\mathcal{L}_{\text{align}}(M_t, A_{t^*})] > \epsilon$$

This captures the fact that artifacts judged aligned at one stage do not guarantee alignment for later, more novel artifacts. Recent reports already suggest this failure mode in agentic systems on the path toward open-endedness. For example, OpenClaw was reported to autonomously schedule a nightly cron job that modified its own configuration file (SOUL.md) by appending new behavioral instructions, with the resulting behavioral drift only noticed two weeks later (CrewClaw, 2026). More broadly, the OpenClaw ecosystem now includes explicit "Self-Evolve" skills that grant agents authority to modify their own configurations, prompts, and memory without user confirmation (Zhang et al., 2026b). These illustrate how alignment may drift over time as the system alters the very mechanisms intended to guide its behavior.

**Alignment of Interactive Components.** OE AI systems are often composed of multiple interacting components, such as agents, models, tools, and evolving environments. Even if individual components are locally aligned, their interactions can give rise to emergent dynamics that are globally misaligned. For example, multi-agent OE systems may converge to equilibria in which harmful behavior emerges despite benign individual incentives. Predicting and constraining such dynamics is particularly challenging due to continual adaptation and co-evolution.

> **R3:** Alignment in OE AI is a distinct problem: values are not specified as objectives, can evolve over time, and emerge from system dynamics.

### 3.4. Traceability

Tracing and reproducing the processes and outcomes of OE AI systems is inherently challenging. The absence of fixed objectives or terminal goals, combined with continual adaptation, makes it difficult to reconstruct how specific

artifacts or behaviors emerge over time. Moreover, small changes in initial conditions or intermediate artifacts can trigger cascading effects, causing the system to diverge rapidly from prior trajectories.

**Lack of Reproducibility.** Reproducing the state of an OE AI system at a given time is substantially harder than for traditional AI systems due to (i) the lack of explicit training objectives and (ii) the difficulty of recreating intermediate environmental feedback and internal states (Flageat and Cully, 2023; Flageat et al., 2024). As a result, it becomes challenging to trace and attribute exploration paths. For example, evolving images that resemble real objects from random initial states is akin to "finding needles in a haystack" (Secretan et al., 2008) in an astronomically large search space. This lack of reproducibility hinders rigorous scientific progress, which relies on transparent, auditable, and repeatable experiments.

**Difficulties in Attribution.** Several methods for oversight and evaluation, such as self-consistency checks (Wang et al., 2023) or interpretability-based tests (Fluri et al., 2024), rely on the ability to compare outcomes across controlled variations. Applying analogous techniques to OE AI is more difficult. Although one can perturb initial conditions to construct counterfactual environments, compounded cascading effects entangle these changes with novelty-driven intermediate dynamics, obscuring causal attribution.

> **R4**: OE AI reduces traceability: specific outcomes are difficult to reproduce or attribute, limiting oversight and diagnostic evaluation.

### 3.5. Resource Wastage

OE AI systems can incur substantial computational and human costs due to prolonged execution and extensive exploration. Unlike traditional ML models that optimize toward specific objectives, OE AI often runs for long durations before producing useful artifacts, if any. For example, discovering a new bin-packing heuristic in FunSearch (Romera-Paredes et al., 2024) required approximately one million iterations to achieve useful result. As the likelihood and timing of valuable outcomes are difficult to predict, the resulting resource expenditure may be difficult to justify. These costs are further amplified when OE AI relies on LLMs, whose scale makes continuous inference particularly expensive. This motivates the need for adaptive resource constraints in OE AI development.

> **R5:** OE AI can consume substantial resources for uncertain or delayed returns.

### 3.6. Social and Human Risks

Beyond technical considerations, OE AI raises broader risks. While all emerging technologies can have societal impacts,

the continual novelty and long-horizon autonomy of OE AI can amplify existing AI harms and introduce qualitatively new ones that are difficult to anticipate or mitigate.

**Rate of Novelty and Loss of Human Agency.** OE AI systems generate novel artifacts at a pace that can exceed society's ability to interpret, integrate, and govern. Rapid, AI-driven innovation risks outpacing institutional adaptation and public understanding, potentially leading to social disruption. Historical precedents, such as the Industrial Revolution, illustrate how accelerated technological change can cause labor displacement and social upheaval. In the context of OE AI, sustained machine-driven discovery may also reduce human agency in shaping scientific and societal progress, leaving individuals increasingly disconnected from processes of creation and decision-making.

**Uninteresting or Misleading Artifacts.** OE AI is expected to generate artifacts that are both novel and useful, yet defining what constitutes "interesting" or valuable progress remains challenging. Models-of-Interestingness (MoI) have been proposed to approximate human judgments of novelty and relevance (Zhang et al., 2024b), but such proxies may be misaligned with human values. As a result, OE systems may produce artifacts that are novel but uninformative, misleading, or narrowly repetitive. The growing complexity of artifacts can further hinder human evaluation, potentially wasting effort and distorting perceptions of progress, or even constraining human creativity by fostering false signals of advancement (Burton et al., 2024).

**Reshaping Human Values.** Adaptive AI systems can influence human beliefs and values over time. LLM-based systems, for example, may learn to mislead evaluators through reward hacking (Wen et al., 2024), creating persuasive but incorrect outputs. In OE AI, cascading effects can amplify such behaviors, leading to the gradual normalization of inaccurate scientific claims, biased policies, or deceptive practices. Prolonged exposure to proliferating artifacts may shift societal norms and values, resulting in a feedback loop where human preferences adapt to machine-generated outputs rather than the system being aligned to human values (Hendrycks et al., 2023).

**Accountability and Responsibility.** Determining accountability for AI behavior is already a challenge for conventional systems, but becomes substantially more complex for OE AI. OE systems act autonomously and produce outcomes that were not explicitly designed or foreseen, complicating the attribution of responsibility. Moreover, OE AI often departs from standard training and deployment pipelines, calling for new legal and institutional frameworks to assign responsibility for harms.

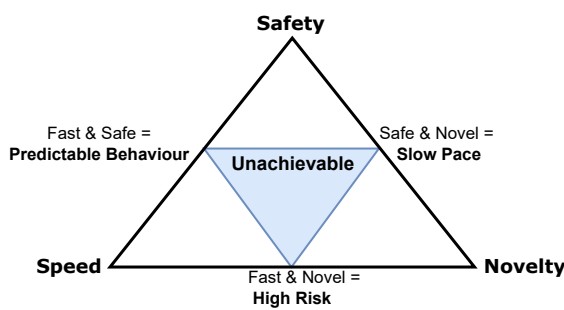

*Figure 1.* The Impossible Triangle of OE AI: safety, speed, and novelty cannot be simultaneously maximized.

---

**R6:** OE AI poses societal risks.

---

### 3.7. Trade-offs

OE AI systems must operate under fundamental trade-offs that constrain their design and deployment. As illustrated in Figure 1, OE AI faces an inherent trilemma between speed, novelty, and safety: improving performance along any two dimensions necessarily degrades the third. Speed denotes the rate at which new artifacts are generated, novelty captures the degree of originality in those artifacts, and safety reflects adherence to constraints and avoidance of harmful outcomes. This trade-off is structural rather than accidental, arising from the core properties of OE exploration. It does not imply that creativity should be broadly limited until safety is fully understood; rather, it means that the balance between creativity, control, and safety should be made explicit rather than left implicit. For example, the OpenClaw ecosystem can be read as an empirical instance of this tension. Within 60 days of launch, OpenClaw reportedly surpassed 300K users, while its community skill registry (ClawHub) expanded to more than 5,400 skills. In parallel, a security audit identified 512 vulnerabilities in ClawHub (Fosters, 2026). This suggests that, in practice, the speed and breadth of open-ended capability growth may outpace the maturation of supporting safety mechanisms, making the study of the trade-off trilemma significant.

**Application-Specific Trade-offs.** At present, there is no standard framework for reasoning about, justifying, or documenting these trade-offs in OE AI systems. In the absence of explicit design-time decisions, systems may implicitly favor speed or novelty, particularly during research-driven development, at the expense of safety. The relative importance of these dimensions depends on the application context, but trade-offs cannot be avoided. In low-stakes domains such as art or games, higher levels of creative exploration may be acceptable even when safety guarantees are weaker. In contrast, in high-stakes settings such as science, robotics, or autonomous systems, exploration may need to be con-

strained more carefully until adequate oversight and safety mechanisms are in place. Similarly, safety-critical domains such as drug discovery or medical diagnosis may require slower exploration and more conservative novelty, while real-time systems such as autonomous driving or industrial control prioritize speed and reliability, constraining novelty to ensure timely and predictable responses. The central concern is therefore not creativity itself, but unconstrained open-ended deployment in settings where failures could be severe or irreversible.

> **R7:** OE AI involves irreducible trade-offs between speed, novelty, and safety, requiring application-specific choices.

## 4. Towards Research for Mitigations

To address the risks identified in the previous section, we outline research directions aimed at mitigating OE AI–specific failure modes as OE AI requires distinct mitigation strategies compared to general agent safety (see Table 2). These directions do not constitute complete solutions, but rather identify technical opportunities for improving oversight, control, alignment, and evaluation as OE systems become more capable and widely deployed.

### 4.1. Oversight

Because the behavior of OE AI systems cannot be fully anticipated in advance (**R1**) and may evolve over long horizons (**R3**), effective oversight during execution is critical. Oversight provides mechanisms to monitor, guide, and intervene in open-ended processes as they unfold, helping to detect unsafe trajectories, prevent cascading failures (**R4**), and maintain alignment with human values (**R3**, **R6**).

**Human-in-the-Loop Oversight.** Humans ultimately define safety and desirable outcomes, making human involvement a central component of OE AI oversight. Human-in-the-loop approaches can include monitoring generated artifacts, filtering which artifacts are propagated to subsequent iterations, or intervening when unsafe behaviors are detected. Human feedback can also be used to steer exploration toward domains deemed valuable or acceptable, as demonstrated in mixed human–AI OE systems (Secretan et al., 2008). However, human oversight is inherently limited by scale and by the difficulty of evaluating complex artifacts, motivating complementary technical mechanisms.

**Interpretable Decision-Making.** Interpretability is a key enabler of oversight, particularly as artifacts become more complex (**R4**). Research directions include requiring OE systems to produce interpretable reasoning traces, for example in natural language, to support failure detection and auditing (Hu and Clune, 2024; Betley et al., 2025). Additional tools, such as feature attribution (Wang et al., 2024b) and analysis of internal representations (Alain and Bengio,

2018; Cunningham et al., 2023), can help overseers understand why specific artifacts or behaviors emerge.

**Hierarchical and Scalable Oversight.** Direct oversight of every artifact is often impractical due to resource constraints (**R5**). Hierarchical oversight structures supervision across multiple layers, in which lightweight monitors screen all outputs and escalate potentially concerning cases to more capable or more costly supervisors (Christiano et al., 2018; Chavan and Chavan, 2024). Scalable oversight methods offer promising approaches for evaluating artifacts that exceed human capacity or appear at high volume, while self-diagnostic techniques (Ibarz et al., 2018; Irving et al., 2018; Kamoi et al., 2024; Huang et al., 2024) can help identify emerging vulnerabilities.

**Adaptive and Co-Evolving Oversight.** Since OE AI systems continually encounter novel and out-of-distribution artifacts (**R1**, **R3**), oversight mechanisms themselves must adapt over time. One research direction is to develop specialized oversight systems—potentially open-ended themselves—that co-evolve alongside the primary OE system and focus explicitly on safety assessment. Such overseers could learn to generalize to new artifact classes and update uncertainty thresholds dynamically as exploration progresses.

**Risk Extrapolation and Constrained Action.** Oversight can also be proactive rather than purely reactive. Specialized OE systems may be used to simulate or extrapolate potential future trajectories of artifacts and assess downstream risks and cascading effects (**R4**). In high-stakes domains, this supports early intervention mechanisms before irreversible harm occurs. Additionally, constraining consequential actions: such as limiting real-world interventions or enforcing sandboxed environments, can reduce risk while oversight and evaluation methods mature. Developing causal models and high-fidelity simulations is a promising direction for enabling controlled exploration (Richens and Everitt, 2024).

### 4.2. Constraints

Most existing safety frameworks assume structured tasks with fixed objectives. In contrast, OE AI systems explore evolving objectives and artifact spaces. Constraints provide a mechanism for shaping exploration without fully specifying goals, allowing designers to manage the trade-offs between novelty, speed, and safety discussed in Section 3.7, while mitigating risks arising from unpredictability and loss of control (**R1**, **R2**, **R7**).

**Constrained Exploration.** OE AI systems often explicitly seek diversity, which can inadvertently drive exploration into unsafe or misaligned regions of the state space. One research direction is to constrain exploration within safety-aware regions, for example by limiting exploration to an $\epsilon$-neighborhood around known safe behaviors, analogous

to safe exploration in reinforcement learning (García and Fernández, 2015). This requires novelty metrics that account not only for difference from past artifacts but also for proximity to safety constraints. In simpler domains, such constraints may be formally specified, while in complex or language-based domains, LLM-based judges could approximate novelty and risk. Techniques such as Gaussian process–based confidence bounds (Sui et al., 2015; Turchetta et al., 2016), reachability analysis (Krakovna et al., 2018; Fisac et al., 2018), or dynamic shielding (Dawood et al., 2024) can be used to reject or penalize unsafe behaviors during exploration. Safety constraints can also be integrated into co-evolutionary frameworks, such as Minimal Criterion Coevolution (Brant and Stanley, 2017).

**Artifact Complexity Budgets.** As OE AI generates increasingly complex artifacts, human oversight and evaluation become more difficult (**R4**). Imposing explicit complexity or novelty budgets can limit the rate or magnitude of change between successive artifacts, helping to balance creative exploration with interpretability and oversight capacity. Such budgets act as safeguards against runaway complexity and reduce the risk of compounding failures over long horizons.

**Rule-Based and Constitutional Constraints.** Although OE AI must adapt to novel situations, there may exist rules or principles that should never be violated. While such constraints cannot capture all possible unsafe behaviors, they can still prevent certain classes of failures. Rules can be specified as high-level principles, such as constitutional constraints (Bai et al., 2022), that remain applicable across changing contexts. LLM-based components may reason explicitly about these rules (Guan et al., 2025) or incorporate causal reasoning to assess compliance (Kıcıman et al., 2024). Recent work (Zaremba et al., 2025) suggests that with sufficient intermediate reasoning, language models can effectively reason about policy compliance, providing a flexible mechanism for enforcing constraints while preserving open-ended exploration.

### 4.3. Adaptive Alignment

Most existing alignment techniques assume a stationary model operating in a relatively fixed environment, allowing safety training to be performed once and then deployed. OE AI violates this assumption: as the system explores new state spaces, acquires new capabilities, and generates novel artifacts, previously learned alignment constraints may become outdated or fail to generalize. This motivates adaptive alignment mechanisms that also evolve (**R3**).

One research direction is continual alignment, in which safety objectives and alignment signals are updated as the system and its operating context change (Zhang et al., 2024a). Prior work has explored dynamic reward weighting (Moskovitz et al., 2024) and adaptive methods for

handling overoptimization and ambiguity (Hong et al., 2024), but these approaches are primarily designed for task-bounded settings and lack mechanisms for sustained feedback over open-ended trajectories. In OE AI, alignment updates must account for continual novelty and a growing diversity of behaviors, rather than incremental deviations around a fixed task.

Adaptive reward and preference modeling offers another promising direction. Techniques such as adaptive preference scaling (Fang et al., 2024; Hong et al., 2024) and distributional preference reward modeling (Li et al., 2024) adjust reward signals in response to shifting human feedback or performance degradation. Extending these approaches to OE AI requires moving beyond scalar reward adjustment toward alignment signals that can generalize across heterogeneous and previously unseen artifacts.

Finally, multi-agent and co-evolutionary formulations provide a natural framework for adaptive alignment in OE systems. Alignment mechanisms themselves can be treated as evolving agents that interact with the primary OE system, enabling continual negotiation between exploration, capability growth, and safety constraints. Such co-evolving alignment dynamics may help mitigate long-term value drift and emergent misalignment in systems whose objectives are not fixed at design time.

### 4.4. Safety Evaluation

Because open-ended AI systems evolve continuously and generate novel artifacts, safety cannot be assessed through one-time testing. Continuous safety evaluation is therefore essential for understanding emerging failure modes, monitoring risk over time, and validating the effectiveness of mitigation strategies.

**Benchmarking OE Safety.** Developing benchmarks tailored to OE AI is critical for quantifying risks that arise from continual adaptation and novelty generation. While existing benchmarks on multi-agent behavior or unintended consequences provide partial insights (Rivera et al., 2020), they do not capture the dynamic and non-stationary nature of OE systems. OE-specific benchmarks should evolve alongside the system, for example, by adapting task difficulty, environmental complexity, or evaluation criteria in response to changing capabilities. Such dynamic benchmarks would enable longitudinal assessment of safety.

**Red Teaming OE AI.** Complementary to benchmarking, red teaming provides a proactive approach to identifying vulnerabilities by actively probing system behavior. For OE AI, red teaming can target individual components or the system as a whole, stress-testing their responses to adversarial conditions, rare events, or pathological exploration trajectories. This may involve manually designed adversarial inputs

or constructed environments that encourage the generation of unsafe artifacts. Recent work shows that LLMs can generate diverse and high-quality artifacts for evolutionary systems (Lehman et al., 2023; Bradley et al., 2023; Liu et al., 2024); similar techniques could be repurposed to generate adversarial artifacts that expose safety failures (Samvelyan et al., 2024). Unlike standard red teaming, the goal here is not to test a fixed model, but to evaluate safety properties of the entire open-ended process as it unfolds.

## 5. Call for Action

Safe development of OE AI requires active engagement from various stakeholders.

**Funding** bodies can shape research priorities. They could urge and dedicate resources to OE researchers to consider and address the safety risks of their work.

**Research** on the intersection of safety and OE research is crucial, impactful, and under-explored. We argue that safety should be a critical part of OE research. Additionally, the AI safety community should dedicate research to the specific risks of OE AI. We hope this paper can provide a bridge to foster exchange and collaboration between these communities.

**Opportunities** lie in the application of OE AI to AI Safety. Aside from providing adaptive oversight (Section 4.1), OE AI can be used to red-team traditional models (Samvelyan et al., 2024) and agentic applications, in addition to automating interpretability research.

**Policy Makers** should mandate audits of sufficiently capable OE AI to ensure adherence to safety standards and societal values.

**Industry** deploying OE AI must implement and rigorously test oversight mechanisms and guardrails for OE systems. Furthermore, a comprehensive evaluation of societal and catastrophic risks should be conducted in collaboration with third-parties, academia and governments.

**Awarness**. Since deploying OE AIs comes with large resource costs and safety risks, the public should be educated and consulted on these decisions to prioritize.

## 6. Alternative Views

While this paper emphasizes the need for proactively addressing the safety of open-ended (OE) AI, several alternative perspectives merit consideration.

**Emergent Cooperation and Self-Regulation.** One view argues that open-ended and multi-agent systems may naturally give rise to cooperative or pro-social behaviors through emergent dynamics, potentially improving safety without explicit constraints or top-down control (Lai et al., 2024; Riedl, 2025). From this perspective, open-endedness is viewed as a mechanism for discovering stable, beneficial equilibria, rather than as a source of risk.
*Our response:* Coordination is not evidence of reliable self-regulation or safety. Coordination observed in controlled settings may not generalize to long-horizon, open-ended, or weakly constrained systems, where collective dynamics can also amplify failures.

**OE as a Tool for Robustness.** Another line of work highlights the benefits of OE methods for improving robustness, for example by generating diverse adversarial examples or stress-testing models during training (Samvelyan et al., 2024; Lehman et al., 2025). Under this view, open-ended exploration is primarily an asset for safety, helping models generalize and resist failure modes.
*Our response:* This is an important positive case for OE methods, but robustness benefits do not remove the need for safety analysis. The same mechanisms that support adaptation and discovery can also produce harder-to-predict failures as systems evolve beyond evaluated regimes.

**Limits of Enforceable Safety.** A more skeptical position is that OE systems are inherently difficult to constrain, and that attempts to impose safety mechanisms may ultimately be ineffective or overly restrictive, undermining the core benefits of OE AI. Relatedly, some argue that excessive caution around OE AI could slow progress toward valuable scientific and societal applications (Chan et al., 2024).
*Our response:* Our position is not that OE development should be broadly constrained, but that proactive safety analysis is necessary for responsible and sustainable deployment. Identifying risks early can preserve the benefits of OE AI while reducing harmful failures and reactive overregulation.

## 7. Conclusion

Open-Ended AI is a promising paradigm for generating novel, adaptive solutions in complex and dynamic environments, driving interest across research and applied domains. However, its open-ended nature introduces specific safety challenges that must be proactively addressed to enable responsible deployment and maximize its societal benefits. We pinpoint unique OE AI risks, which differ from those of task-bounded AI systems, and highlight the importance of human and automated oversight. Further, we suggest ways of providing adaptive guidelines to OE AI that retain its creativity while co-evolving with the system. Lastly, we call for targeted, continuous safety evaluations and provide concrete suggestions on how different stakeholders can contribute to the responsible development of OE AI. Ultimately, we hope this paper will lead the OE and safety communities, as well as other stakeholders, to treat safety as a central consideration in the development and deployment of OE AI.

## Impact Statement

This paper aims to urge the OE and safety community to consider safety as a priority when developing OE AI. As such, it will reduce potential harms from OE AI in the real world while allowing for larger beneficial adoption of OE AI. While prioritizing safety might slow down the progress of OE AI, we argue that it is essential to have safe-by-design OE AI from first principles to proactively avoid risks before they happen.

## Acknowledgements

The authors would like to thank Joel Lehman for the insightful discussion on the paper. This work was partially funded by ELSA – European Lighthouse on Secure and Safe AI funded by the European Union under grant agreement No. 101070617. The work was also partly funded by the German Federal Ministry of Education and Research under the funding code 16KIS2012 (AIgenCY). The responsibility for the content of this publication lies with the authors. Views and opinions expressed are, however, those of the authors only and do not necessarily reflect those of the European Union or European Commission. Neither the European Union nor the European Commission can be held responsible for them.

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

| Dimension | General Agent / AI Safety | Open-Ended (OE) AI Safety |
|---|---|---|
| Objective structure | Fixed or explicitly specified objective (e.g., reward or loss) | No fixed objective; goals emerge from open-ended processes |
| Temporal scope | Finite-horizon or episodic behavior | Long-horizon, potentially unbounded evolution |
| Predictability | Partial predictability under known objectives | Structural unpredictability due to continual novelty |
| Evaluation | One-time or periodic evaluation | Continuous, longitudinal evaluation required |
| Alignment target | Aligning a policy to a reward or preference model | Aligning an evolving process and its dynamics |
| Distribution shift | Typically bounded or task-specific | Endogenous and unbounded state-space expansion |
| Failure modes | Specification gaming, reward hacking, robustness failures | Emergent misalignment, value drift, cascading failures |
| Control mechanisms | Static constraints and post-training safeguards | Constraints must adapt as system and environment evolve |
| Traceability | Decisions and outcomes often reproducible | Outcomes difficult to reproduce or attribute |
| Resource profile | Predictable training and inference costs | Speculative, long-running, resource-intensive exploration |

*Table 1.* Comparison of safety challenges in general agent/AI systems versus open-ended (OE) AI systems.

| Safety Lever | General Agent / AI Safety | Open-Ended (OE) AI Requirements |
|---|---|---|
| Oversight | Human or model-based review of outputs | Continuous, hierarchical, and adaptive oversight |
| Alignment methods | Static reward or preference modeling | Adaptive alignment that evolves with the system |
| Constraints | Fixed rules, filters, or action constraints | Dynamic, context-aware constraints shaping exploration |
| Evaluation | Static benchmarks and test suites | Dynamic benchmarks that co-evolve with capabilities |
| Red teaming | Testing a fixed model | Red teaming the entire open-ended process |
| Interpretability | Explaining individual decisions | Tracing evolving behaviors and trajectories |
| Governance hooks | Deployment-time safeguards | Design-time trade-off specification and monitoring |
| Resource control | Budgeting training and inference | Adaptive resource budgets under uncertainty |
| Failure response | Rollback or retraining | Early intervention and trajectory abortion |
| Scope of guarantees | Point-in-time safety guarantees | Process-level risk management over long horizons |

*Table 2.* Comparison of mitigation strategies for general agent/AI safety and open-ended (OE) AI safety.

