# OpenReview forum: "Position: Safety Must Precede the Deployment of Open-Ended AI Agents"
_ICML.cc/2026/Position_Paper_Track — ICML 2026 Position Paper Track regular_

### Official Review · Reviewer_VEpY · 2026-03-12

**Significance:** 3
**Argument Clarity:** 3
**Rating:** 5
**Confidence:** 4

**Questions:**

1. Could the authors clarify what would count as OEAI evaluation protocol? Maybe proposing a more concrete one would be helpful.
2. The impossible triangle is interesting. Could the authors provide more evidence/results/justification on that? It is a little bit conceptual for an important claim throughout the paper.

**Alternative Views Section:**

No

**Compliance With Llm Reviewing Policy A Conservative:**

Affirmed.

**Discussion Potential:**

3

**Final Justification:**

The rebuttal addressed my concerns. I have increased my score from 4 to 5.

**Paper Summary:**

This paper argues that safety considerations should precede the large-scale deployment of open-ended AI systems. The authors claim that systems designed for continual exploration and novelty generation introduce new safety challenges. The paper also discusses these risks and outlines potential research directions for oversight, alignment, and evaluation to ensure the responsible development of open-ended AI. However, I didn't find the mandatory *Alternative Views* section in the content, although the authors did discuss similar issues in Section 4.

**Position:**

Yes

**Position In Title:**

Yes

**Related Work:**

3

**Strengths And Weaknesses:**

Strengths
1. The paper presents an important perspective on open-ended AI. It is being discussed in different fields such as agentic systems and scientific discovery, etc. Safety measures are really important in that sense.
2. The stated position is also clearly articulated early and maintained consistently in the paper. The authors argue that the defining properties of OE AI introduce a qualitatively distinct safety problem that should be examined before large-scale deployment.
3. The paper is well-written and easy to follow.

Weaknesses
1. I think some of the core risks are not specifically discussed. For example, unpredictability, misalignment, and loss of control are important, but in practice they can refer to many different phenomena. From my perspective, although the paper proposes the concept and definition of novelty, but the risk claims are not as formal as that. I would say it would be better if the safety claims are also formalized in a way.
2. I think there is also some imbalance between the breadth of the concerns raised and the depth of support for each one. It might be better for a position paper to narrow down into less but more specific claims and analyze them deeper.

**Support:**

3

---

> ### Author Rebuttal · Authors · 2026-03-30
>
> We would like to thank the reviewer for acknowledge that our paper presens important safety perspetc, with well written and articulated position in the paper.
>
> We would like to note that Section 6 is the **Alternative Views** section where we present 3 different views along with our rebuttal. We are happy to rename Section 6 to Alternative Views. Please find the answers to your questions below:
>
> ---
>
> > Q1: Could the authors clarify what would count as OEAI evaluation protocol?
>
> We thank the reviewer for the suggestion. We would like to connect this to section 4.4. In particular, we emphasize that evaluation in OE AI should focus on the evolving process rather than a static model snapshot, and include (1) longitudinal evaluation over trajectories of generated artifacts, (2) dynamic benchmarks that adapt to the system’s capabilities, (3) intervention tests that evaluate whether oversight mechanisms can detect and halt unpredictable, unsafe trajectories (R1). We have update this in section 4.4 and refer the reader in L273 to this.
>
> > Q2: The impossible triangle is interesting. Could the authors provide more evidence/results/justification on that?
>
> We thank the reviewer for finding the impossible triangle interesting. We proposed the impossible triangle as a conceptual framework to highlight a recurring design tension in open-ended systems: increasing novelty typically reduces predictability and therefore makes safety assurance harder, while adding stronger safety controls or validation tends to slow exploration and artifact generation. In addition to L269-272, we will strengthen the justification. We will clarify that the triangle is a conceptual trilemma grounded in the structural properties of OE systems, rather than a mathematically proven impossibility result. As mentioned by Reviewer 3gj9, we hope that this paradigm sparks a discussion building upon the applications dependent trade-offs mentioned in L250-L261.
>
> > W1: I think some of the core risks are not specifically discussed. For example, unpredictability, misalignment, and loss of control are important, but in practice they can refer to many different phenomena.
>
> We thank the reviewer for this suggestion. We agree that the safety claims should be formalized more explicitly, not only the notion of novelty. In the current draft, unpredictability is tied to the OE definition through the observer’s prediction loss (L110-120), and our arguments on loss of control are built on from Ecoffet et al., 2020. R3:Misalignment is decomposed into evolving-system misalignment and interaction-driven misalignment (L187-205).
>  We note that R3 is not derivable from novelty and learnability alone in the same way as R1, because misalignment requires an additional notion of human values or alignment evaluation. We are happy to add this formalization to the revision to better ground R3 in the same framework as the OE definition. Due to space limits, please refer to our formalisation as our rebuttal to **Reviewer 2-6G8U (Q)**.
>
>
> > W2: I think there is also some imbalance between the breadth of the concerns raised and the depth of support for each one. It might be better for a position paper to narrow down into less but more specific claims and analyze them deeper.
>
> We thank the reviewer for this suggestion. Our goal in this position paper was to provide a framing of the safety issues raised by recent open-ended AI systems, particularly those increasingly built on LLM-based and agentic architectures. Because the area is still emerging, we chose first to map out the main categories of risk and argue why OE AI warrants dedicated safety attention, rather than focus narrowly on only one or two concerns. While prior works have examined safety issues and failure cases in systems that lie along the continuum toward OE AI, our aim here is to provide a unifying framework for discussing these risks and failure modes.
> We agree that each of these risks could be developed in greater depth, and that a narrower treatment would allow more detailed analysis of individual claims. However, doing so would substantially change the scope of the paper. Our intent here is to provide an initial taxonomy and research agenda that future work can build on and refine.
>
> ---
>
> We thank the reviewer for their careful review. We will include the above discussions in the paper. We would be happy to address any further questions the reviewer may further have. We hope the paper and rebuttal contributes to a broader and important discussion of OE AI and its safety implications.

---

> > ### Author Rebuttal · Reviewer_VEpY · 2026-04-01
> >
> > My concerns have been resolved so I will increase my score to 5.

---

### Official Review · Reviewer_3gj9 · 2026-03-12

**Significance:** 3
**Argument Clarity:** 3
**Rating:** 5
**Confidence:** 3

**Questions:**

as above

**Alternative Views Section:**

Yes

**Compliance With Llm Reviewing Policy A Conservative:**

Affirmed.

**Discussion Potential:**

3

**Final Justification:**

My concerns have been addressed.

**Paper Summary:**

This paper argues that open-ended (OE) AI, defined as systems that continually generate artifacts that are novel and learnable to an observer, introduces a distinct class of safety challenges not adequately addressed by existing AI safety frameworks. The paper formalizes novelty and learnability (Section 2.1; Equations 1 and 2), then articulates seven risk themes R1–R7, including intrinsic unpredictability, creativity–control trade-offs, misalignment in evolving systems, reduced traceability, resource wastage, societal risks, and an asserted “impossible triangle” of speed, novelty, and safety (Figure 1). It proposes research directions for oversight, constraints, adaptive alignment, and continuous safety evaluation (Section 4), and concludes with a call for coordinated action by funders, researchers, policymakers, and industry (Section 5).

**Position:**

Yes

**Position In Title:**

Yes

**Related Work:**

3

**Strengths And Weaknesses:**

Strengths


1. The position is explicitly stated on Page 2 (“Position: While OE AI offers substantial potential benefits, it also poses unique and significant risks that must be addressed preemptively...”). This clarity persists throughout Sections 3–5, consistently arguing for OE-specific safety work prior to broad deployment.
Organization and argument flow:

2. The progression from definition (Section 2) to risk analysis (Section 3) to mitigations (Section 4) is coherent. Risks are neatly enumerated as R1–R7, which aids readability and anchors downstream proposals.


3. Section 2.1 gives a formalization of novelty and learnability (Equations 1 and 2). Even if imperfect (see weaknesses), making the assumptions explicit is constructive for a position paper that aims to catalyze a safety conversation.
Useful comparative summaries:

4. Table 1 contrasts general AI/agent safety with OE AI safety across ten dimensions. This helps position OE AI as a qualitatively different regime for risk management.
Table 2 maps typical safety levers to additional OE requirements (e.g., continuous, hierarchical oversight; co-evolving benchmarks). This provides a practical mental model for practitioners thinking about adapting existing tools.
Potential for community discussion:

5. The asserted "impossible triangle" of speed, novelty, and safety (Figure 1) is provocative and likely to spur discussion, even among skeptics. It invites concrete debate on trade-off quantification and governance documentation.


weakness:

1. Minor presentation/typo issues that impair precision. Several references miss full bibliographic detail or links are placeholders ("[LINK]").

2. "Broader Implications" outlines alternative perspectives but treats them briefly, with limited references or detailed rebuttals. For instance, the "self-regulation via emergent cooperation" view is summarized with one citation.

3. Sparse empirical or case-based support for the safety claims. The resource wastage example (Section 3.5) cites FunSearch iterations (Page 4), and other sections rely on high-level reasoning. No case studies of OE incidents, red-team outcomes, or oversight breakdowns are offered.

4. Limited specificity in proposed mitigations relative to OE particulars. These are sensible, but many are adaptations of general safety techniques. Clearer articulation of what is uniquely required for OE (e.g., novelty-aware gating functions, trajectory-level rollback semantics, longitudinal risk accounting) would elevate the section.

**Support:**

3

---

> ### Author Rebuttal · Authors · 2026-03-31
>
> We would like to thank the reviewer for their careful review of the paper. We are pleased that the reviewer found the paper clear and coherent, and that Table 1, Table 2, and the impossible triangle may help stimulate community discussion on trade-offs and governance. We respond to the specific comments below.
>
> ---
> >W1: Minor presentation/typo issues that impair precision.
>
> We thank the reviewer for pointing out that 2 of our references were printed with [Link] holder due to a format mismatch. We have updated this.
> > W2: "Broader Implications" outlines alternative perspectives but treats them briefly, with limited references or detailed rebuttals.
>
> Thank you for your comment. In the Broader Implication/Alternative views section, we included three main alternative arguments along with our rebuttal to them in Section 6. Due to the lack of space, we could only dedicate half a column to the alternative views. Additionally, we could not find many citations for the alternative view. Instead of having our rebuttal at the end we are happy to include the rebuttal for each alternative view discussed. Regarding the citations, below are the additional citations that we can include. Please let us know if the reviewer had a particular paper which we may have missed and we are happy to include it.
>
> AV 1: We will add [1]  showing that multi-agent LLM systems can exhibit higher-order coordination and complementary collective behavior under particular prompting conditions. However, we clarify that evidence of coordination is not yet evidence of reliable self-regulation or safety across settings, especially for long-horizon or unconstrained OE systems.
>
> [1] Emergent coordination in multi-agent language models.
>
> AV 2: We will add [2], arguing that evolutionary/open-world approaches may address forms of uncertainty that standard ML handles poorly. We agree this is an important positive case for OE methods; we respond that robustness benefits do not eliminate the need for safety analysis, because the same mechanisms can also produce harder-to-predict failures.
> [2] Evolution and the knightian blindspot of machine learning.
>
> AV 3:  We added a citation from the broader AI governance literature on regulation–innovation tradeoffs. We clarify that our position is not that OE development should be broadly constrained, but that proactive safety work is a condition for responsible and sustainable deployment.
>
> [3] Balancing the tradeoff between regulation and innovation for artificial intelligence: An analysis of top-down command and control and bottom-up self-regulatory approaches."
> > W3:Sparse empirical or case-based support for the safety claims.
>
> We appreciate the shared observation that the paper would benefit from empirical evidence. **Since submission**, several incidents and empirical studies have emerged that provide this grounding and an early glimpse of risks we identify. While we still don't have OE systems in the formal definition, we increasingly have autonomous agents that perform more open ended tasks while adapting their design and harness. This strengthens our position that identifying risk categories is important before large scale deployment. As also mentioned by all of the reviewers, we hope this leads to wider discussion on safety of OE preemptively.  Due to char limit, we kindly point the reviewer to our rebuttal to **Reviewer 2-6G8U (W)**.
> >W4: Limited specificity in proposed mitigations relative to OE particulars. These are sensible, but many are adaptations of general safety techniques.
>
> We are glad that the reviewer found our OE risks section detailed. In the appendix, we include Table 1 and Table 2, which compare safety challenges and mitigation strategies in general agent/AI safety versus OE AI safety. Our point is not that OE safety requires an entirely separate toolbox, but rather that familiar methods must be adapted to handle continual novelty, long-horizon adaptation, and process-level rather than snapshot-level risk, also acknowledged by the reviewer in S4. This includes novelty-aware gating or constraint functions (Sec 4.2) that regulate exploration as new artifacts emerge, trajectory-level intervention rather than only post hoc filtering, and longitudinal risk accounting that tracks how safety properties change over time. With an additional page in the camera-ready version, we would be happy to move this discussion from Appendix into the main paper, which would improve readability and better contextualize the proposed mitigations. We thank the reviewer for this suggestion.
>
> ---
> We thank the reviewer again for their supportive comments that improved the paper. We will include the above discussions in the paper. We are happy to address any further questions the reviewer may further have. We hope the paper and rebuttal contributes to a broader and important discussion of OE AI and its safety implications.

---

> > ### Author Rebuttal · Reviewer_3gj9 · 2026-04-03
> >
> > Thanks for the response. My concerns have been resolved, and I will adjust the score accordingly.

---

### Official Review · Reviewer_6G8U · 2026-03-13

**Significance:** 4
**Argument Clarity:** 3
**Rating:** 5
**Confidence:** 3

**Questions:**

Is it possible to derive R3 (alignment) from the formal definitions of OE, instead of using only analogy to biological evolution

**Alternative Views Section:**

Yes

**Compliance With Llm Reviewing Policy A Conservative:**

Affirmed.

**Discussion Potential:**

4

**Paper Summary:**

The paper studies the safety of open-ended (OE) AI systems. It argues that such OE AI systems have different kinds of challenges than task-bounded or static models. Also, existing AI safety frameworks designed for fixed-objective systems are not sufficient for OE systems.
The paper concludes with a call to action and provides comparison tables for OE safety vs general AI safety.

**Position:**

Yes

**Position In Title:**

Yes

**Related Work:**

3

**Strengths And Weaknesses:**

***Strengths***

1. The paper addresses the safety issues of OE AI systems which is not covered in the AI safety literature whose focus was mostly about fixed-objective AI systems. This is very timely and relevant.

2. The formal definitions ground the positions of the paper

3. The conceptual contribution of the trade-offs between  speed, novelty, and safety.

***Weaknesses***

The paper could benefit from empirical evidence for the risk factories, e.g., misalignment and social risks.

**Support:**

3

---

> ### Author Rebuttal · Authors · 2026-03-31
>
> We thank the reviewer for their encouraging review of our paper. We are glad that the reviewer finds our paper timely and grounded. We appreciate the reviewer’s recognition of our safety trilemma. Please find answers to your questions below:
>
> ---
> > Q: Is it possible to derive R3 (alignment) from the formal definitions of OE, instead of using only analogy to biological evolution
>
> We thank the reviewer for the suggestion. We can introduce the definition of R3 using the loss-based definition of OE in addition to the analogy to biological evolution. We note that R3 is not derivable from novelty and learnability alone in the same way as R1, because misalignment requires an additional notion of human values or alignment evaluation. We formalise it as below:
>
> Let \mathcal{L}_{align}(M_t, A_{t'}). denote the alignment loss assigned by the observer model $M_t$ to a future artifact $A_{t'}$, where lower values indicate better alignment with human values. We say that an artifact $A_{t'}$ is certified as acceptably aligned at time $t$ if
>
> \mathbb{E}[\mathcal{L}_{align}(M_t, A_{t'})] \le \epsilon
>
> for some threshold $\epsilon$. We can then formalize the failure of temporal alignment certification as
>
> $$
> \exists t<t'<t^* :
> \mathbb{E}[\mathcal{L}_{align}(M_t, A_{t'})] \le \epsilon
> \;\land\;
> \mathbb{E}[\mathcal{L}_{align}(M_t, A_{t^*})] > \epsilon.
> $$
>
> This suggests that alignment evaluations performed at time $t$ may hold for nearer-horizon future artifacts but fail for later ones as the OE system continues to evolve and generate increasingly novel outputs. We are happy to add this formalization to the revision to better ground R3 in the same framework as the OE definition.
>
> >W: The paper could benefit from empirical evidence for the risk factories, e.g., misalignment and social risks.
>
> We appreciate the shared observation that the paper would benefit from concrete, real-world evidence. **Since submission**, several incidents and empirical studies have emerged that provide this grounding and an early glimpse of risks we identify. While we still don't have OE systems in the formal definition, we increasingly have autonomous agents that perform more OE tasks while adapting their design and harness. This strengthens our position that identifying risk categories is important before large deployments. As also mentioned by all of the reviewers, we hope this leads to discussion on preemptive OE safety.  We will incorporate the following, mapped to our risk taxonomy:
>
> R1 / R4: In Feb 2026, a public documentation [1] of an OpenClaw agent that autonomously started deleting the email inbox of the user. The cause was not adversarial. The agent's context compaction mechanism discarded the initial safety instruction ("confirm before acting") when working memory became saturated. It continued operating under task objective, and the user was unable to halt it remotely. This demonstrates how OE agents operating over long horizons can lose safety-relevant state through their own autonomous resource management, with the constraint violation being neither predictable nor traceable in real time.
>
> R2 / R3: [2] reported their OpenClaw agent autonomously scheduling a nightly cron job to modify its own configuration file (SOUL.md), appending new behavioral instructions to itself. It was noticed only two weeks later after observable behavioral drift. The OpenClaw ecosystem also now includes explicit "Self-Evolve" skills granting agents full authority to modify their own configs, prompts, and memory without user confirmation [3].
>
> R3: Moltbook, a social platform exclusively for AI agents (launched Jan 2026, 100K+ agents), has produced empirical evidence of emergent behavioral properties at scale. The MoltNet study [4] found that agents drift away from their stated personas over time.
>
> R5 / R7: The OpenClaw ecosystem's rapid scaling provides indirect evidence for the speed–novelty–safety tension. Within 60 days of launch, OpenClaw surpassed 300K+ users; its community skill registry (ClawHub) grew to 5,400+ skills. However, a security audit found 512 vulnerabilities on ClawHub [5]. This trajectory where the speed of OE capability growth outpaces safety infrastructure is an empirical instantiation of the trade-off we formalize in the impossible triangle.
>
> Further EU Machinery Regulations consider fully or partially self-evolving behaviour as requiring stricter conformity assessment.
>
> [1] She runs AI safety at Meta. Her AI agent still went rogue, The SF Standard
>
> [2] openclaw-agent-permissions-safety , crewclaw blog
>
> [3] MemRL: Self-Evolving Agents via Runtime Reinforcement Learning on Episodic Memory
>
> [4] MoltNet: Understanding Social Behavior of AI Agents in the Agent-Native MoltBook
>
> [5] Don’t get pinched: the OpenClaw vulnerabilities, Tom Fosters
>
> ---
>
> We would like to thank the reviewer for their efforts in reviewing and strengthing our paper. We look forward to further discussion that will contribute to engagement with OE AI safety.

---

> > ### Author Rebuttal · Reviewer_6G8U · 2026-04-04
> >
> > I thank the authors for addressing my questions

---

### Official Review · Reviewer_KvSz · 2026-03-15

**Significance:** 4
**Argument Clarity:** 2
**Rating:** 4
**Confidence:** 4

**Questions:**

I didn’t fully understand $M_t$ and $A_t$. $M_t$ is a model and what is it predicting?  Do you want this perhaps -->  $L(M_t(A_{1:t}), A_t)$?  and is M_t predicting any future $\hat{A}_{t’}$? Also what is expectation over?

It's unclear if the paper was focusing on safety again autonomous evolving of the LLM or safety again delibrate intent, like read teaming. I see safety defined as ability to tackle the former but also red teaming was mentioned in the later part of the paper.

Are the artifacts $A_t$ are only the novel artifacts that are evolving over time or all the artifacts? I was thinking all initially but in the speed part it looks like only the novel ones are considered.

The relation to constraint exploration and dynamical system is intetesting!

**Alternative Views Section:**

No

**Compliance With Llm Reviewing Policy A Conservative:**

Affirmed.

**Discussion Potential:**

4

**Paper Summary:**

The paper argues for open-ended syste which they defined similar to Hughes et al. (2024) as systems that autonomously generate novel, learnable artifacts for an external observers. The OE AI introduces a distinct and underexplored class of safety challenges that differ qualitatively from those of task-bounded or static AI systems.
It presents the following position: "While OE AI offers substantial potential benefits, it also poses unique and significant risks that must be addressed preemptively before further development. Its inherent unpredictability and long-horizon dynamics necessitate dedicated safety research."
The authors identified 7 risks that the open ended AI may poses: R1: Unpredictability, R2: Creativity vs. Control, R3: Misalignment, R4: Traceability, R5: Resource Wastage, R6: Social and Human Risks, R7: Trade-offs (Impossible Triangle).
 Then the paper briefly touches upon research towards mitigating these risks and calls for action.

**Position:**

Yes

**Position In Title:**

Yes

**Related Work:**

3

**Strengths And Weaknesses:**

Strengths:
- Safety is critical and this paper presents a good risk taxanomy to generate discussions around the topic
- Touches upon the relevant research directions to mitigate the risks
- Tries to mathematically define open-ended ness

Weakness:
- Mathematical formulation is not concrete: I didn’t fully understand $M_t$ and $A_t$. $M_t$ is a model and what is it predicting?  Do you want this perhaps -->  $L(M_t(A_{1:t}), A_t)$?  and is M_t predicting any future $\hat{A}_{t’}$? Also what is expectation over?
- Missing the relation of these formulation to safety aspects. I think if safety can also be linked to the above definition that would provide a better framework for the paper
- I find paper to be vague and verbose at multiple instances, specially when defining risks. I think if they could provide concerete examples of what could go wrong will support their position
- R2 says creativity requires sacrificing control; and section 4 proposes constraints and their position is preceeding safety before development. Is the position also arguing to limit cretivity until safety mechanism are understood?
- I founds the mitigation section also provide only vague measures without concrete proposals, prioritization, or feasibility analysis. It's hard to know what to actually do after reading the paper.
- Adding the arugments which go against this position will also be helpful

**Support:**

3

---

> ### Author Rebuttal · Authors · 2026-03-30
>
> We would like to thank the reviewer for acknowledging the safety position and appreciating the risk taxonomy. We are pleased that they consider that our paper will generate discussion potential among the community. We thank the reviewer for appreciating the relation to constraint exploration and dynamical systems.
>
> Re W6: We would like to note that Section 6 is the **Alternative Views** section, where we present 3 different views along with our rebuttal. We will rename the section. Please find the answers to your questions below:
>
> ---
> > Q1/W1: I didn’t fully understand $M_t$ and $A_t$. $M_t$ is a model and what is it predicting?
>
> We thank the reviewer for their comment. We intended that $M_t$ denotes the observer's predictive model after observing the history $A_{1:t}$, rather than a model that separately takes $A_{1:t}$ as input each time. In the revision, we clarify this explicitly by defining
> $$
> M_t := \mathrm{Obs}(A_{1:t}),
> $$
> where $\mathrm{Obs}$ denotes the observer's update rule. $M_t$ induces a predictive distribution over future artifacts,
> $$
> P_{M_t}(A_{t'} \mid A_{1:t}),
> $$
> for $t' > t$. We also specify that $\mathcal{L}(M_t, A_{t'})$ measures prediction error on a realized future artifact, for example via negative log-likelihood. The expectation is taken over the randomness of the artifact-generating process $S$ (and observer randomness when applicable). We agree that this makes the formalization much clearer.
> >Q2: Unclear see safety defined as ability to tackle the former but also red teaming was mentioned in the later part of the paper.
>
> The paper primarily focuses on safety risks from the autonomous and evolving behavior of OE systems themselves, rather than deliberate malicious intent. Red teaming is included later not as the core threat model, but as an evaluation method for stress-testing such systems and uncovering failure modes, including under adversarial conditions. We will clarify this distinction explicitly in the revision.
>
> > Q3: Are the artifacts $A_t$ are only the novel artifacts that are evolving over time or all the artifacts? I was thinking all initially but in the speed part it looks like only the novel ones are considered.
>
> Here, $A_t$ denotes the artifact generated at time $t$; the sequence $A_{1:t}$ includes all generated artifacts. Novelty is defined as a property of this sequence through the observer’s increasing prediction difficulty over time.
>
> > W2: Missing the relation of these formulation to safety aspects. I think if safety can also be linked to the above definition that would provide a better framework for the paper
>
> We defined our definition of safety in Section 2.2. In terms of formalisation of risks, we have included some formal justifications in unpredictability and now also for misalignment (please see our response to R2-6G8U).
>
> > W3/W5: Vagueness
>
> We have Tables 1 and 2 to more clearly distinguish OE-specific risks and mitigations from those in general AI safety, thereby reducing vagueness in the overall framing. We included some examples, the FunSearch case to illustrate the speculative and resource-intensive nature of OE search. More broadly, our position is intentionally forward-looking: the paper aims to identify and clarify risks before large-scale failures from fully open-ended systems occur, rather than only after such incidents have been observed. However since the submission, we have observed obvious risks of systems that may not be OE but lie on the continuum with major safety risks. Please see our response to **Reviewer 2-6G8U (W)**.
>
> Following the reviewer’s comment, we plan to move Tables 1 and 2 from the appendix into the main paper to improve readability and make these distinctions more explicit.
>
> > W4: Is the position also arguing to limit cretivity until safety mechanism are understood?
>
> We thank the reviewer for raising this important point. Our position is not that creativity should be broadly limited until all safety mechanisms are fully understood. Rather, we claim that open-ended AI introduces an inherent trade-off between creativity, control, and safety, and that this trade-off should be made explicit rather than left implicit. In low-stakes domains, such as art or games, higher levels of creative exploration may be acceptable even with weaker safety guarantees. In contrast, in high-stakes settings, such as science, robotics, or autonomous systems, it may be necessary to constrain exploration more carefully until adequate oversight and safety mechanisms are in place. Our argument is therefore not against creativity itself, but against unconstrained open-ended deployment in settings where failures could be severe or irreversible.
>
> ---
> We thank the reviewer for their supportive comments and detailed feedback. We will include the above discussions in the paper. We are happy to address any further questions the reviewer may have. We hope that the rebuttal contributes to the broader discussion of OE AI and its safety implications.

---

> > ### Author Rebuttal · Reviewer_KvSz · 2026-04-08
> >
> > Thank you for the response.
> >
> > > $M_t$ denotes the observer's predictive model
> >
> > I would suggest to not use model for $M_t$ in this case.
> > I think authors could spend time in defining input and output spaces where $M_t$ is used for more clarity.  M_t is used at 3 places, output of observation model, input of loss function and as parameters to define predictive distribution. Is space of all same?
> >
> > > Q3
> >
> > Sorry i forgot, how was speed defined? Is it $A_{1:t}/t$ or only $novel(A_{1:t})/t$?
> >
> > > W2
> >
> > Sorry, i am still missing how these formulation are linked to safety aspects?
> >
> > > Our argument is therefore not against creativity itself, but against unconstrained open-ended deployment in settings where failures could be severe or irreversible.
> >
> > But you also argue that there is a trafeoff between the safety and novelty.
> > This may be wrong message though, i would sugges to use the work exploration instead of novelty?

---

### Decision · Program_Chairs · 2026-04-30

**Decision:**

Accept (regular)

**Comment:**

This paper argues that the safety of Open-Ended AI systems is an important topic that requires dedicated research.
This is a reasonable position with important consequences. The reviewers are all positive about this work: 2x Accept (5) and 2x Borderline Accept (4).
They also agree about its Significance (2x excellence and 2x good) and the Discussion Potential (2x excellence and 2x good).

Some of the weaknesses aspects of this work, mentioned by the reviewers, are:
- Mathematical formulation is not very clear.
- The Alternative Perspectives (mentioned within the Broader Implications section) is brief.
- Limited specificity in proposed mitigations.

The authors' rebuttal addressed at least some of these issues. As a result, three of the reviewers increased their scores.
Overall, I believe the topic is probably not controversial, but is important. I recommend acceptance of the paper.